# A Survey of the Landscape Visibility Analysis Tools and Technical Improvements

**DOI:** 10.3390/ijerph20031788

**Published:** 2023-01-18

**Authors:** Zhiqiang Wu, Yuankai Wang, Wei Gan, Yixuan Zou, Wen Dong, Shiqi Zhou, Mo Wang

**Affiliations:** 1College of Architecture and Urban Planning, Tongji University, Shanghai 200092, China; 2Bartlett School of Architecture, University College London, London WC1H0QB, UK; 3Tongji Architectural Design (Group) Co., Ltd., Shanghai 200092, China; 4Shanghai Tongji Urban Planning & Design Institute Co., Ltd., Shanghai 200092, China; 5College of Design and Innovation, Tongji University, Shanghai 200093, China; 6College of Architecture and Urban Planning, Guangzhou University, Guangzhou 510006, China

**Keywords:** landscape, visible green, 3D Isovist, spatial configuration, built environment

## Abstract

Visual perception of the urban landscape in a city is complex and dynamic, and it is largely influenced by human vision and the dynamic spatial layout of the attractions. In return, landscape visibility not only affects how people interact with the environment but also promotes regional values and urban resilience. The development of visibility has evolved, and the digital landscape visibility analysis method allows urban researchers to redefine visible space and better quantify human perceptions and observations of the landscape space. In this paper, we first reviewed and compared the theoretical results and measurement tools for spatial visual perception and compared the value of the analytical methods and tools for landscape visualization in multiple dimensions on the principal of urban planning (e.g., complex environment, computational scalability, and interactive intervention between computation and built environment). We found that most of the research was examined in a static environment using simple viewpoints, which can hardly explain the actual complexity and dynamic superposition of the landscape perceptual effect in an urban environment. Thus, those methods cannot effectively solve actual urban planning issues. Aiming at this demand, we proposed a workflow optimization and developed a responsive cross-scale and multilandscape object 3D visibility analysis method, forming our analysis model for testing on the study case. By combining the multilandscape batch scanning method with a refined voxel model, it can be adapted for large-scale complex dynamic urban visual problems. As a result, we obtained accurate spatial visibility calculations that can be conducted across scales from the macro to micro, with large external mountain landscapes and small internal open spaces. Our verified approach not only has a good performance in the analysis of complex visibility problems (e.g., we defined the two most influential spatial variables to maintain good street-based landscape visibility) but also the high efficiency of spatial interventions (e.g., where the four recommended interventions were the most valuable), realizing the improvement of intelligent landscape evaluations and interventions for urban spatial quality and resilience.

## 1. Introduction

The role of urban landscape visibility is increasingly being studied and discussed due to the growth of the urban design discipline. First, landscape visibility has always been a key factor in the evolution of an urban system. However, the potential value of human visual perception has not been fully studied. Only in the last several decades have researchers become aware of people’s spatial exploration behavior and visual cognition largely linked with the changeable landscape visual reachability [1]. In return, the visuospatial cognition of a place can activate the dynamics of urban space [2,3,4,5,6]. The importance of view analysis and the terms of the analysis method, such as “visual absorption”, “visual connections”, “visual corridor”, or “visual intrusion”, were stipulated by Lynch [7]. Second, to some extent, the value of the visual environment is as important as quantitative indicators such as accessibility. The activity behaviors driven by people’s landscape perception of a site will have a long-term impact on regional mobility, as well as the production and living space of the city [8,9], or even become a sense of a region [7]. Visible green spaces can largely extend the sensorial green corridor connection, enabling cities to provide more visually resilient spaces [10]. Thus, a reasonable visual landscape distribution can greatly promote regional value. Third, the visual field is the primary way humans connect to their surroundings [11]; thus, visibility analysis could be a common lens to study human–landscape connections [12].

However, because of the dynamic nature of people’s movement and what they perceive when interacting with the environment (including the visual background and visual attractions or landscapes), it is difficult to find a measurement tool that adequately models all of the complex situations [13]. The interaction between humans and space is complex and dynamic [7]. It is difficult to define a feeling before it is quantified [14,15]. For example, in Chinese gardens, there are more artistic ways of expressing the visual feeling of the environment. For example, the scenery moves with the viewers, there is a change of scene at every step, and winding paths lead to a peaceful place and a wide margin of enlightenment. These descriptions are extremely subjective, and there is no way to accurately describe the space quantitatively from the perspective of visual feelings, neither can they be systematically methodicalized or instrumented. Most of the traditional types of research cannot be carried out; therefore, quantifiable scientific indicators are needed to serve designers in a real sense to better understand the relationship between space and landscape visual qualities [16].

In light of this, we reviewed the analysis methods and tools through the decades, comparing the advantages and disadvantages of the tools and finding the common deficiencies when they are used in current urban planning and design through many related studies. We found that there are few effective tools to analyze different landscape visuals in complex environments in a superposition state, resulting in scientific results from existing tools that do not fit well in the actual visual perception of the landscape that people can receive in a dynamic environment. Secondly, and more importantly, the existing analysis lacks scientific suggestions to continue optimizing the urban planning scheme and can only be used as a reference for designers, which has been disregarded by many relevant studies.

In this paper, we proposed a possible working path to optimize urban space based on the existing line-of-sight landscape visibility analysis methods. Our innovations include the following: (1) We proposed an urban ISO-matric-scan method based on standardized voxelized units to record the landscape visibility and blocking for each building surface corresponding to a certain scenario. In addition, this standardized voxel-based data recording approach can enable us to accumulate multiple scenarios of landscape visibility with the same unit of measurement for analysis, thus making it possible to analyze the complex dynamic visible landscape of cities. (2) We extracted and calculated multiple spatial variables of buildings in the context of big data and machine learning by constructing machine learning models to discover which building variables have a determining influence on optimally enhancing the corresponding complex landscape visibility. (3) We also used the spatial relational model to more accurately determine the magnitude of the impact of different building variables on the dynamic landscape visibility of the entire site within a local geographic location. These innovations provide a scientific foundation to support designers in knowing precisely how and where to control building variables with minimal intervention to achieve more visible green landscapes in complex scenarios. 

## 2. A Review of Approaches to Landscape Visibility Analysis

### 2.1. Definitions and Generations of Previous Methods

The measurement of landscape visual quality developed rapidly after a series of analytical theories were proposed. Around 1967, the digital analysis of landscape visibility was proposed by Tandy [17] and Amidon and Elsner [18]. The term “viewshed”, introduced by Tandy [17], is defined as a geographical region visible from a location, and this was the basis for the first digital visibility measurement concept. This research led to the development of a multitude of methods for the quantitative analysis of space perception. Subsequently, the Isovists method was proposed by Benedikt, in which he introduced a set of analytic measurements of Isovist properties [19]. Isovist redefined the space and divided the environment through the configuration of different visual fields. This insight provided the potential for bridging the interaction between human vision and urban configuration in subsequent studies.

Subsequently, spatial visibility was explored in spatial quantification. Batty [20] and O’Sullivan and Turner [5,21] adopted the node calculation method based on the concept of Isovists, which can be conducted in the program to measure the overall visibility in a geometric space and is suitable for macro and micro scenes.

Based on this method, the raster and grid methods were extended for more concise quantitative analyses, which have become the mainstream methods for visual space computing. The calculations were optimized and extended from 2D to 3D by exploring the visualscape [22], 3D visual space [23], viewsphere [24], and other methods proposed by subsequent researchers, as shown in Table 1.

These analysis methods are very representative and valuable for further research. However, fewer of them have been examined in an actual urban environment rather than in an independent experimental space. For example, Suleiman, Joliveau, and Favier [41] studied an enclosed area within 0.8 km^2^. We found the common shortcomings (Table 1) when applied to current urban projects, which can be summarized as follow:

First, early methods were more concerned with visualization rather than quantitative measurement [24,47]. Even if, in a handful of studies, the results can be quantified, they are still difficult to calculate in large urban environments (i.e., most of the urban planning studies focused on small (local) extents, while the natural landscape studies were cross-scale extents) [12]. There is a gap in the complexity and scale of the environment.

Second, almost all of the methods were discussed from a simple viewpoint, while the environment in which people behave is complex. The computing power of the existing tools for complex urban environments is still insufficient.

Moreover, few of the previous studies mention that the impact of the mobility of cities matters in visual observation problems. Only Morello and Ratti [34], Leduc and Kontovourkis [48], and Fisher-Gewirtzman [49] tested the sequential visual perception of streets and connected it with pedestrian movement behavior through visibility and mobility. Chang and Park [43] and Putra, Yang, and Yang [50] combined the analysis results with machine learning models for validation. More importantly, none of the studies used scientific evaluation data to precisely indicate how spaces should be modified to meet better environmental visibility. Therefore, these methods are difficult for use as supporting techniques in actual planning.

To bridge this gap, we used the batch computing method and proposed our ISO-MATRIC-Scan method, i.e., a batch analysis process for storing the computational results of multiple scenarios overlaid by a changeable resolution when computing visibility. Through the algorithm of the loop batch scanning model volume, the quantized values from multipoint, line, and surface attractions or observation points can be accumulated and recorded on the building surface or building voxel units.

### 2.2. Technical Comparison

Following this line, we classified the optimization process of the visibility analysis techniques into four main generations according to their power of measurement, scientificity, and potential analytical value for urban problems.

Generation 1 (2D measurement)—The use of a single viewpoint to connect with any geometric node in the observed area. The nodes are calculated to delineate the visible and invisible areas as long as the line of sight could be directly connected to the nodes without being blocked. The results are presented as a visualization reference, with only limited parameters being exported, such as for 2D Isovist [17,19] and 2D VGA [20,21].

The great progress in upgrading the analytical power from 2D to 3D in visibility analysis techniques is related to the development of related software, such as GIS, which brought computational power to support this promotion.

Generation 2 (3D geometry metric computation)—The geometric edges that the line of sight does not intersect are stretched on the 3D space to materialize the visible and invisible spaces. The vertical calculation dimension is added based on the two-dimensional visibility in Generation 1, which is closer to the measurement of the real visual perception of people and is more suitable for the visibility analysis of buildings with different heights in an actual city (e.g., 3D view [23] and 3D viewsphere [24]).

Generation 3 (3D raster/mesh)—The grid surface of the building block hit by the line of sight is divided into the visible and invisible, and the calculation accuracy can be adjusted according to the grid size. Because the grids are standard and can be quantified, the results have the potential for further study rather than just a presented visualization, (e.g., 3D Isovist [34,51] and 3D visibility VGA [38,46]).

Generation 4 (3D voxel based)—Voxels hit by the line of sight can be counted as visible and invisible. The added value is that the building surface calculation can only calculate the current nearest block face once at a time but not the block face behind. The voxel calculation method can calculate all of the blocking spaces simultaneously (e.g., 3D Isovist advanced [35,36] and 3D voxel [25,37]).

From a technical point of view, we summarized the main shortcomings of the previous generations of analysis methods; the common weakness of the earlier analysis methods of Generations 1–3 was that they were not intelligent and responsive enough. They are concerned more with the static analysis of existing results, which cannot take feedback as the supporting parameters of the design optimization through the interaction between the measurement and solution in the design and planning process.

Furthermore, the experimental methods in Generations 1–4 are mainly measuring the objects from a simple preset viewpoint, which is difficult to switch between the observation point and the observed landscape.

To bridge this gap, we proposed Generation 5 to combine the advantages of standard grid computing, voxel-based variable accuracy, and interventional feedback, and we added batch cycle scanning and recorded the workflow, which makes it possible to solve the problems of line, plane, and superposition visibility in complex landscape problems and, consequently, visual mobility and cross-scale issues.

### 2.3. Strengths and Weaknesses of the Existing Tools

To date, there are many 3D view analysis tools used to study urban space and landscape issues, and several typical tools are listed below (Table 2 and Table 3).

(1)ArcGIS and ArcGIS Pro

ESRI introduced visibility analysis tools (viewshed) in the ArcGIS software around 2000. The algorithm was based on the theory of viewshed [17], which can determine the visible range of mountains and valleys for solving the landscape visibility of the terrain. The tool was originally designed to analyze terrain and conduct watershed analyses, and the algorithm was initially designed for one single viewpoint and has low computational scalability when the number of viewpoints increases [52]. The application of the tool was mainly concentrated on regional landscape and ecological problems, rather than on urban and architectural issues [53,54]. In subsequent iterations, the viewshed algorithms were improved by Lee and Stucky [55]; Kim, Rana, and Wise [56]; Domingo-Santos et al. [47]; and others [54]. In the latest ArcGIS Pro version, the viewshed algorithm is optimized, and multiple viewshed viewpoints can be set.

(2)Supermap

The Supermap company inherited and optimized the GIS viewshed analysis tools in the ArcGIS software of the ESRI GIS platform. The analysis module of the point-to-point visual corridor and skyline analysis module based on the viewnets or line-of-sight [57] method were also added. This tool was mainly aimed at the visual range of space and natural landforms in the city to perform more intuitive operations, and it can also analyze complex urban spaces. The use of the tools is conducted at a fast calculation speed, with an adjustable calculation accuracy [58]. It is a method of visualization analysis, which can split the urban space into visible and invisible parts. The quantitative problem in the planning research has yet to be solved, and the analysis results still lack specific parameter outputs [58].

(3)Depth Space 3D based on Graph Theory’s Depthmap

In 2020, Opoarch developed the Depth Space 3D software by using the pace syntax theory, extending the analysis methods of Isovist [19] and VGA MAP [21] to 3D VGA. Complex urban spaces, such as a building street and green spaces, can be more precisely computed [46]. This tool is mainly aimed at the visual range of space and natural landforms in the city to perform more intuitive operations, and it can also analyze complex urban spaces. The use of the tools is conducted at a fast calculation speed, with an adjustable calculation accuracy. Sophisticated calculations can be made for complex urban spaces, such as environments containing many trees [46]. Relevant applications of the software are still yet to be examined, since many modules are still in the demo version.

(4)3D Isovist of the Grasshopper plug-in DeCodingSpaces and a view analysis of the Grasshopper plug-in Ladybug

Around 2018, Robert McNeel & Associates and the ETH Future City Lab developed a plug-in, DeCodingSpaces, for Grasshopper. In addition, Isovist [19] and VGA MAP [21] were introduced and upgraded to the 3D Isovist algorithm function, which can perform fine 3D visual analyses and operations on complex geometric building forms, streets, and other small-scale indoor spaces. The calculation results can be exported as vector geometry results and parametric results [59]. The application of this tool is mainly concentrated on building interiors and small-scale street spaces [60]. The results are provided in the form of parameters and have a large potential to facilitate further processing, while the calculation speed is slow, which is hardly suitable for large-scale urban spaces and natural environments [60].

The Ladybug tool was developed by Mostapha, in 2012 and the first version was released in January 2013. The methods of viewshed [17], Isovist [19], VGA map [21], etc., were added to the modules view analysis and view windrose, as well as other plug-in functions. The view analysis module can perform fine 3D visual analysis operations and sky visibility calculations on complex geometric building forms, streets, and other middle-scale urban spaces. The computational results are also quantified. This tool can be widely used in building spaces, street spaces, large-scale urban spaces, and on natural landforms, while the accuracy of the analysis can be adjusted to meet the needs of different environmental scales. The results can be exported in the form of lists of the parameters for further processing. Since the tool is open access, users are allowed to be involved in the refinement of its functionality over time (Table 2 and Table 3).

**Table 2 ijerph-20-01788-t002:** Theoretical support for each tool. The color highlight can better illustrate the performance level, with light grey is mediocre, dark grey is good, and white is poor.

Supported and Related Methods	ESRI Series ArcGIS; GRASSGIS	Supermap	Depth Space 3D	Rhino Grasshopper Plugin DeCodingSpaces	Rhino Grasshopper Plugin Ladybug
Viewshed [17]	Optimized	Optimized			Simplified
Isovist [19]			Optimized	Usage	Usage
VGA [21]			Optimized	Simplified	Optimized
Voxel [36]					
Visualscape [22]	Simplified				
Viewscape [61]	Simplified				
Viewnets or Line of Sight [57]	Optimized	Optimized		Optimized	
3D Visibility/Ray Casting [40]	Optimized	Optimized			Optimized
3D VGA [44]			Optimized		

**Table 3 ijerph-20-01788-t003:** Comparison of the mainstream tools for visibility analysis. The color highlight can better illustrate the performance level, with light grey is mediocre, dark grey is good, and white is poor.

Performance Comparison	ESRI Series ArcGIS; GRASSGIS	Supermap	Depth Space 3D	Rhino Grasshopper Plugin DeCodingSpaces	Rhino Grasshopper Plugin Ladybug
User Interaction					
Viewpoint and Analysis Object Selection	Layer selection	Click	Click, Layer selection	Click, Region selection	Click, Region selection
Visibility Obstacles					
Building Obstacles	Good	Good	Good	Good	Good
Terrain Obstacles	Good	Good	Good	Good	Good
Tree Obstacles	Mediocre	Mediocre	Mediocre		
Data source					
Building Data	Mediocre	Good	Good	Good	Good
Landscape Terrain Data	Good	Good	Good	Mediocre	Good
Urban surface Data	Mediocre	Mediocre	Good	Mediocre	Good
Open Source					
API Acquire	Fully opensource				Fully opensource
Computing Capacity					
GPU-Based Parallel Processing			Unknown	Usage	Usage
Parallel Multiple Viewpoints Computing	Mediocre	Mediocre	Good	Mediocre	Good
Analysis Object					
DEM/Raster	Usage	Usage	Usage	Transfer	Transfer
Shape/Polygon	Transfer	Transfer	Transfer	Usage	Usage
Mesh			Transfer	Usage	Usage
Brep				Usage	Usage
Airborne LiDAR		Usage			
Outcome					
Visualization	Good	Good	Good	Good	Good
Parametric	-	-	-	Measure the list of line of sight	Measure the list of mesh values, measure the distance from VP to the mesh centroid
Geometric	Raster	-	Triangular polygon surface	Result Isovist geometry, line of sight	Result triangular Mesh, Mesh centroid
User Intervention	Low, Fixed result	Low, Fixed result	Low, Fixed result	Varies with applications	High, Evaluation of outcomes
Method Generation	2–3 ***	2 **	2–3 ***	2 **	3 ***

Notes: **, ***stand for analysis performance level 2, 3 respectively.

At the same time, a large number of researchers have conducted case studies in different regions (Table 4). For example, Puspitasari and Kwon [62], based on the above 3D ISOVIST method, conducted an empirical study on local landmark high-rise buildings in Pudong, Qingdao Shinan, and Mega Kuningan, and performed a multiregional quantitative comparison. Yin [63] analyzed and compared the skyline of multiple nodes on a street in Buffalo based on the 3D viewshed and line-of-sight methods. Morais et al. [45] performed a visibility analysis on a relatively complex landscape area in Oporto based on the 3D VGA method, and Leduc and Kontovourkis [48] linked visibility and mobility through pedestrian movement behavior.

These collected studies conducted empirical case studies. However, these case studies have certain limitations and can hardly perform overall landscape performance evaluations of urban areas. First, these studies were mainly based on the observation and analysis of simple viewpoints; for example, the object of analysis was only landmark high-rise buildings [62]. Second, the large-scale urban environment analysis was not realized. Most of these case studies were street-based analyses or within a 1 km^2^ urban region (Table 4). Third, the suggested intervention of space was the missing part in all of the evaluations, which is insufficient for guiding the subsequent urban design or urban spatial renewal.

Therefore, a technical tool to evaluate not only the visibility of space but also the possible solutions based on the quantitative analysis is desired in urban planning. In addition, the simple viewpoint as the study hypothesis is not enough to simulate the actual situation and the dynamic flow in the complex situations.

## 3. Visual to Perceptual: Towards Dynamic Landscape Perceptions

### 3.1. Optimization Route and Conceptual Framework

We proposed an optimization method (Figure 1) based on the Ladybug plug-in, which mainly focuses on the following improvements and innovations.

First, we broke down the problem of simple viewpoints with small-scale urban environments. (1) The current research analysis related to visibility was based on a single target or single viewpoint, which was relatively hard to adapt to multiview scenes and cross-scale scene analysis. (2) These analyses are often generalized small-scale analyses of a single viewpoint with simple analysis variables (e.g., openness and visibility), lacking purposeful analysis of specific landscapes and landmarks, or global applicability of a large urban environment [12]. (3) A handful of studies have the potential to link visibility analysis with more attributes. Hence, the analysis results were difficult in terms of providing practical assistance to urban planners and designers.

In this regard, we improved the data type of the viewpoint for the calculations and made it possible to run grid calculations with a changeable accuracy, regardless of whether the observables and observation points are single, linear, or surface domain. At the same time, it is easy to switch the analysis scenarios and objects for different scenarios, computational accuracies, and speeds according to the task’s objectives, and we ensured the credibility and uniformity of the quantitative results in the computation of cross-scale, cross-object, and complex superposition problems (Figure 1).

Second, the existing methods generally lack endogenous conversion based on the spatial visual viewing performance; therefore, most of the current studies using these tools can only stay at the analysis level rather than optimizing the morphology through the visual quality. Meanwhile, there has been less mention of quantifiable evaluation methods, which constitutes good urban spaces under the visual evaluation criteria.

We optimized the output not only as a visual representation but to correlate it to standardized architectural voxels, thus promoting a greater potential for subsequent analyses and interventions. The output can thus be used to accurately and dynamically evaluate the spatial by using machine learning and spatial indices. Based on such improvements, the output evaluation results are not only an independent indicator but have the potential to be deeply combined with the analysis of multiple factors, such as the actual overall urban development value, architectural conservation, and human activities.

Third, we expanded the use of machine learning and spatial indices to accurately and dynamically evaluate the spatial viewability and used OLS regression to find correlations between the urban spatial form variables, as a suggested means to optimize the landscape view control. On the other hand, we used spatial regression models to spatially discover the strength of the influence of the different morphological variables to suggest the most spatially efficient enhancement areas and the certain recommended means of urban morphological interventions (Figure 1).

Therefore, through the architecture of this new analytical process, we bring the analysis tools from small-scale experiments conducted in the laboratory to a more urban scale and urban design application scenarios. It also serves as a bridge between traditional physical space simulation analyses and statistical model construction under the new perspective of the era of big data [64,65]. Our method solves the current problem that most of the research mentioned above only remained at the level of analysis and evaluation and lacked clear design guidelines for interventions that can be implemented. Thus, this method can provide urban designers with a new tool for optimizing interventions in urban space and architecture with reliable data-based evaluations.

### 3.2. Performance Examination Criteria

To compare the value of the improvements, the rubrics for the potential of the analysis were set in terms of the computational performance, complexity of the scenario, correlatability of the site landscape values and other indicators, responsiveness of the interaction with the tool, and value of the proactive suggestions for planning and design. We tested and studied these dimensions on our case study in Shaoxing.

Among them were the (1) computational performance, which mainly included the scope of the test site, size of the accuracy employed, and computation time; (2) the complexity of the scenario, which mainly included multiview problems, cross-scale problems, and cross-object problems (i.e., streets, open spaces, and buildings). We took a site area smaller than 10 square km as the local-scale and 10 to 1000 square km as the landscape scale [12,66,67]. (3) Whether a comprehensive quantification and evaluability of the landscape value of the site could be easily and quickly calculated by outputting variable factors and having a reasonable computational medium. (4) Whether it is possible to proactively recommend a clear scope of implementation and the means of intervention based on the evaluation value output to the planning designer so that the perceived value of the landscape can be optimized in the planning and design.

## 4. Data and Methods

### 4.1. Study Area

We focused on the central section of the Shaoxing Ancient Conservation Area (10.9 km^2^) as a case to test our empirical hypothesis. Located at 119°53′02″–121°13′38″ E and 29°13′36″–30°16′17″ N. Shaoxing is situated south of the mouth of the Qiantang River, one of the oldest capitals and most dynamic humanistic life scenes in the Yangtze River Delta Metropolitan Region [68].

As a historical city with ancient city-style features, Shaoxing has many humanistic street attractions, natural mountains, water bodies, sites, and buildings with diversity and richness; therefore, Shaoxing was used as the test site to meet our needs for complex scenario testing. Secondly, tourism, as a very important urban function in Shaoxing, makes urban imageability particularly important, and a landscape view analysis can be used as a good analytical dimension for potential urban spatial value enhancement.

In the case of the Shaoxing Ancient Conservation Area, due to the rapid expansion of the city, a large volume of construction and development are densely covered, blocking the view of the corridor of the natural landscape and weakening the ancient pattern of the city [69,70,71]. Shaoxing plans to renovate this unreasonable urban configuration by scientific means and reconstruct the main streets conforming to the context of the urban landscape [72,73]. With the help of our proposed large-scale dynamic visual analysis method, we provide technical support for planners and guides on where and how to intervene in the space more effectively.

### 4.2. Data

There are four clusters in the dataset used in this research: (1) a building shapefile to calculate the visibility and the urban spatial attributes for correlation computation; (2) the topography data constructed using a digital elevation model (DEM); (3) a road network shapefile to situate the possible evaluation background for proxying the landscape view received by the pedestrian flow; (4) a pedestrian flow heatmap using the Baidu API (http://api.map.baidu.com/Ibsapi/, accessed on 11 August 2022) to capture the popular areas of activity. The first two variables involve data on the environmental base plate of this study, and the third and fourth clusters in the dataset serve as inputs for the proxy street-based evaluation.

#### 4.2.1. Spatial Background

The building and road data were obtained through the Amap’s API (2022, https://lbs.amap.com/, accessed on 17 May 2022) to obtain a total of 8139 polygon building data and 2993 polyline road data for 2019 [71]. The topographic data were loaded into the Rhino software by reloading the DEM 30 m accuracy elevation topographic map with a 10 m × 10 m grid (Figure 2).

#### 4.2.2. Viewpoint Input Settings

The viewpoint settings were divided into two steps to more realistically represent the viewing possibilities under different scenarios. By following the Baidu heatmap, we first identified four major streets with the highest density of pedestrian and vehicular traffic, which were used as street-level inputs for the evaluation of the visual value of major streets. We divided the road network into equal 50 m spaced points as street-based viewpoint input data.

Then, considering that the interoperability of sightlines dictates that places, where landscape attractions can be seen, can also be seen by the position where attractions are located (Figure 3a). The three major mountains, including Fu mountain, Ji mountain, and Kuaiji mountain, were selected as the mediums for attractions A, B, and C, respectively, for the landscape visibility evaluation. The set of points obtained by dividing the contour lines by a 10 m height spacing and a horizontal 30 m spacing was used as the input for the landscape inspection of the major hills.

#### 4.2.3. Visibility and Spatial Indices Measurement

A series of calculations were performed on the geometric building forms to determine the six basic architectural spatial parameters (Table 5) that can be intervened in the physical space and the visibility and visibility blocking measures (Table 5).

(1) Density represents the density of buildings within a 100 m radius of the selected building unit; (2) height measures the relative height of the building unit above ground level; (3) geometries length measures the perimeter of the base of each building unit; (4) angle deviation with 90 degree measures the 90 degrees minus the angle between the long side of each building form’s envelope rectangle and the view direction to represent the angle between the building’s main structural orientation and the viewpoint; (5) geometries projection length measures the actual projection length of the long side of the building’s envelope rectangle relative to the viewpoint; (6) distance from landscape focus point measures the linear distance of each building to the landscape focus point; (7) visibility measures the visual score of each building matrix surface and ground surface in multiple scenarios. For example, first, the measurement from the landscape attractions measures how much visibility the entire homogeneous site will receive from the main body of the landscape. Second, the measurement from the main streets measures the actual visibility received by pedestrians with the highest heat density from the entire urban space. Thus, the deviation in the landscape visibility of the test site by pedestrians on the most active streets can be obtained by calculating the absolute value of the difference in the visibility values on the building facade between the landscape visibility measurements and the street-based measurements, which can be formulated using Equation (1), as follows:(1)$$Visibility\;deviation_{j}=\left|Visibility_{street}-Visibility_{j}\;\right|$$(8) Visibility blocking measures, for the most popular streets, how much the visibility of the landscape component is blocked by specific buildings. As the blocking can be affected by the visible building facade area and the distance attenuation, we borrowed from the principle of optical imaging and set Equation (2) as:(2)$$Visibility\;blocking_{\left(i,j\right)}=\sum_{j=1}^{j}Visible\;facade\;area_{i}\times Visibility\;deviation_{ij}/Dis\_Pt_{\left(i,j\right)}\;$$
where the $Visible\;facade\;area_{i}$ measures the visible area of building *i*, and $Dis\_Pt_{\left(i,r\right)}$ is the distance from building *i* to the landscape attraction *j*.

A core area of 10.9 square kilometers was selected to comprehensively discuss the landscape value of the three mountains that can be perceived on the street pavement and building facade.

First, the whole urban space was divided into 20 m × 20 m standard matrices in batches to serve as the standard plate of the evaluation data. Second, the analysis view attractions were sampled by picking the points of the contour line on the reconstructed terrain at an interval of a 30 m as the landscape attractions for the analysis. Third, the geometries were selected for the case study area within a 10.9 km^2^ terrain zone in total and 8139 buildings for the occlusion calculation. Fourth, the observations of the movement of the dynamic flow were proxied using 50 m equipartition points on the street pavement of four main streets, namely, Zhongxing Road, Jiefang Road, Shengli Road, and Renmin Road. Lastly, the visibility values of the multiple landscape viewpoints on each building were recorded as the superposition results on each 20 m building matrix (Figure 3b). We used a loop in Grasshopper to record each calculation result.

With the visibility and blocking calculations, we wrote the results of all three landscape scenario calculations into each equal-area building raster and accumulated the results of the three scenarios to obtain an overall integrated landscape visibility value (Figure 3c). To further increase the computational speed, we accumulated all of the measured variables on each building matrix into a 100 m radius hexagonal grid for resampling as the data for subsequent machine learning, and we obtained a total of 1823 data and 1682 nonzero data.

### 4.3. Visibility-Oriented Interaction Model Architecture

We hypothesized that variables such as the density of the building and its surrounding buildings, height, length of the bottom side, the angle between the long side of the building and the view, length of the projection of the long side of the building to the view direction, and the distance relationship between the building and the view attraction point all affect its visibility and blockage of the street view. These spatial variables basically constitute the factors that can influence the shape of the building block. We constructed an OLS model to test the explanatory power of the different variables on the visibility of this area to represent their effectiveness as spatial variables to intervene in the urban form.

These urban building form variables were used in this study to test the validity and supportability of our landscape visibility-led spatial value evaluation and intervention process for urban design and, in the future, these variables can be extended to any data that can be sampled on the building surface units (e.g., building color, building year, type, material, function, and time-of-use data in the temporal dimension).

Lastly, multicollinearity might exist among the building spatial variables, since some of them tended to be closely correlated. The Pearson correlation test was used to examine variables with correlation problems (pair plot value > 0.8; VIF value > 10). Before running the OLS and GWR models, we removed the variables that were unimportant and achieved significant multicollinearity with other important factors, namely, sum height, geo_len (i.e., average and sum geometries lengths), sum area, and all visibility (Figure 4, Table 6). Then, we fixed our hypothesis with Y (Visibility) = X1 (Density) + X2 (Height) + X3 (proj_len) + X4 (90De_angle).

Through the correlation learning in the OLS among the multiple parameters (Figure 5), the most influential architectural variables that affected the regional landscape visual perception could be found, making it possible to effectively intervene and test the effectiveness.

## 5. Results and Discussions

### 5.1. Landscape Spatial Value Evaluation

Through the OLS analysis, we found that in the case of Shaoxing, the building height, the building long side projection distance, and the angle between the building and the viewing direction all significantly influenced the visibility of the three hills. We also found that the long side projection distance of the building was the most central variable in reducing the overall landscape quality of the three hills, with a correlation coefficient of 0.7365, followed by a building height of 0.3375 and an angle between the building and the viewing direction of 0.2833 (Table 7).

Therefore, some specific interventions can be proposed in the subsequent urban renewal and urban landscape appearance control. When faced with maximizing the street-level visibility of the three mountains, the planning designer can prioritize maximizing the visibility of the mountains by controlling the overall building projection length, followed by controlling the building height and the angle of the building orientation to the visual center of the landscape.

### 5.2. Influential Obstacles and Interventions

In order to obtain better recommendations for interventions that can be practically implemented, we also took the spatial variability into account in the model. Based on the OLS variables model, we used GWR to observe both the overall variables’ impact on the landscape visibility and the explanatory strength of these variables at local geographic locations, so that for each area we can provide a spatial variable that has the most significant impact on the landscape visibility degradation and, thus, we can target the modification of the case area. We spatially observed the local R square distribution of each of the four variables (i.e., density, height. Proj_len, and 90De_angle) (Figure 6, Table 8).

We ranked the four local R square values that could be sampled for each building to obtain the most central influential variables for the different areas such that the most effective interventions for urban landscape control could be spatially visualized (Figure 7).

In this study, a multivariable correlation evaluation was carried out for each building, and through the correlation with multiview blocking, the buildings that were most in need of intervention were found. Moreover, considering the coefficient of the different variables, the corresponding intervention means, such as the height reduction, building width reduction, rotation, and density reduction, are given for each suitable area. In detail, we found that 946 buildings need to control their height, 1230 buildings need to limit their projected facade length, 176 buildings require the control of their orientation, and 3249 buildings are located in areas with an excessive overall density. These findings can support the project towards a more environmentally integrated urbanscape through a human–machine hybrid enhanced decision-making process.

### 5.3. Test of the Predefined Goals

This study not only analyzed and summarized the existing landscape visibility analysis tools but also proposed improvement methods to address some of the limitations in practical application scenarios. Through this case study, we tested our proposed data-driven visibility analysis process, and through the process structure and analysis, we obtained data-supported design tools that can be implemented to improve the overall landscape visibility. First, it provides clearer guidance for urban planning and design than traditional visibility analysis. Second, although in this case only four form-space-based variables were considered, the analysis method has great potential to expand these variables to provide recommendations for more complex scenarios. Third, in setting the analysis scenarios, we can continuously overlay the view assessment of different scenarios through data logging to obtain a more complex and comprehensive result for regression analysis, which allows the analyst to participate and control the analysis assessment of each scenario at any time, with good interactivity.

### 5.4. Limitations

First, the problem of shading the actual view using green trees in the analysis also needs to be considered, and the method in this study, if trees are taken into account in the modeling, will somewhat reduce the computational performance, and it is also difficult to obtain accurate information regarding the location and volume of trees (i.e., it is difficult to efficiently calculate the spatial linkage with the model by field measurement modeling or by identifying trees through street view images).

Second, the degree of the distance attenuation in the landscape perception generated by the actual perspective was ignored in this study, and it is difficult to provide a scientific basis with supporting data for how the effect of this variable interferes with the human perception of the environment.

Third, the question of how to define the position of the landscape value is also an issue that needs to be discussed in depth; whether the more landscape that is seen, the greater the value and resilience, or whether only the local landscape value is retained to maximize the socioeconomically functional benefits of the differentiated regional development. This question cannot be adequately discussed in our existing proposed analysis.

In addition, we must of course be careful to note that landscape visibility is only part of the intuitive factors in the regional value, and various factors affect the landmarks, landscapes, and interesting attractiveness of an area, among which the historical culture, function behind the form, historical precipitation of the area development, etc., may affect people’s perception of the value of an area’s landscape and the benefits it can bring to this place.

## 6. Conclusions

In this paper, we first presented a new approach to record the dynamic landscape visibility of complex scenarios in every voxel of a building surface by applying a batch-processed ISO-matric-scan calculation, which allows for the spatial variables of each building on the site to be re-evaluated in terms of their different impacts on the overall landscape visibility through landscape visibility in both global and local perspectives.

Second, through the empirical study of Shaoxing, we obtained clear guidance on how and where to improve the built environment. This, in turn, can more directly support designers with a scientific basis for the adjustment of the space. For example, the analysis precisely suggests that designers can intervene with the height in 946 of the highest-impact buildings and with the projected facade length in 1230 of the highest-impact buildings, resulting in a renewed urban space with a better publicly perceived green landscape.

Third, our study benefits from big data and machine learning analysis workflows that go beyond human-understandable intuitive three-dimensional spatial correlations to provide a fuzzier dynamic evaluation that encompasses the precise local to global domain. The study also provides a new approach that uses controllable simple spatial variables to interfere with the complex green environmental quality of the overall site.

Therefore, it is grounded that our approach can fill the gap in the relevant research field on how to provide guiding spatial improvement suggestions based on a complex visible landscape. This method also has the potential to continue to be investigated in the field of urban resilience.

## Figures and Tables

**Figure 1 ijerph-20-01788-f001:**
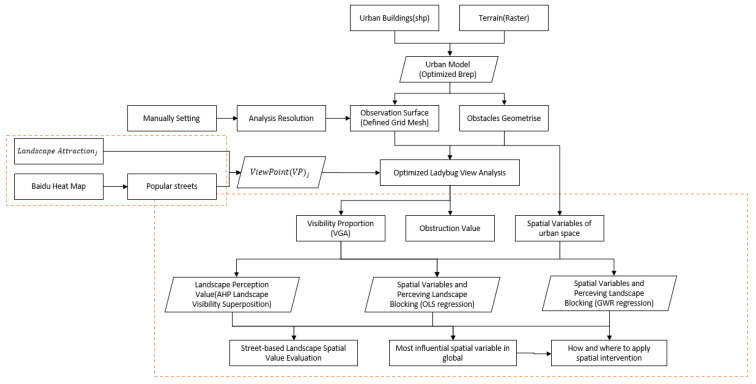
Conceptual framework of the new optimized spatial visual analysis route.

**Figure 2 ijerph-20-01788-f002:**
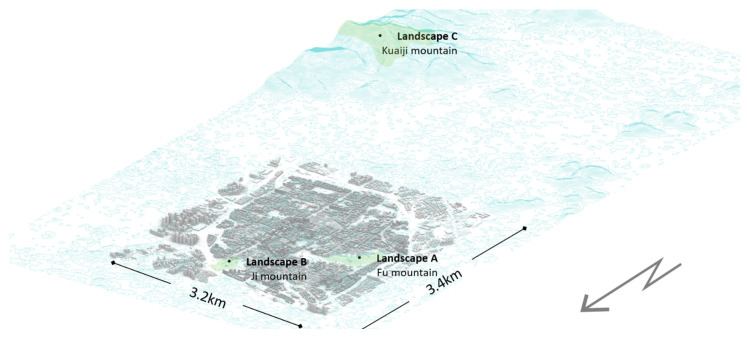
Presentation of spatial data of the test site.

**Figure 3 ijerph-20-01788-f003:**
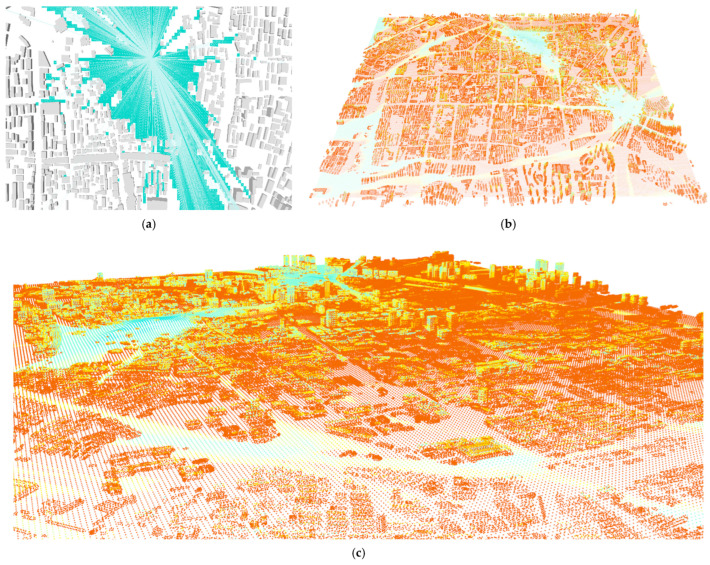
Multiple landscape attractions visibility: (**a**) demonstration of 3D ISOVIST by an optimized Ladybug view analysis; (**b**) data record by an ISO-matric-scan for multiple attractions; (**c**) data record by an ISO-matric-scan for multiple attractions, using a 20 by 20 m grid.

**Figure 4 ijerph-20-01788-f004:**
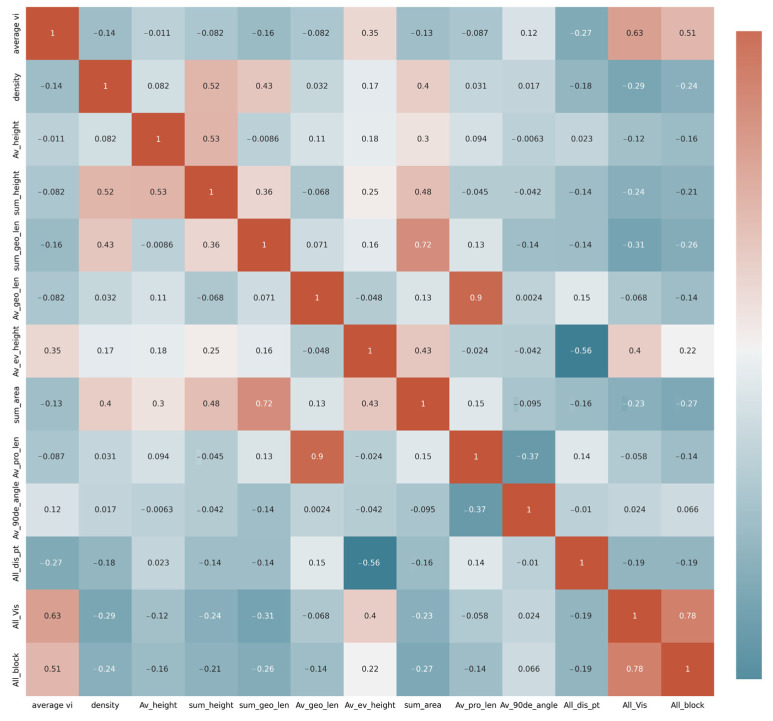
Pearson correlation of the selected variables.

**Figure 5 ijerph-20-01788-f005:**
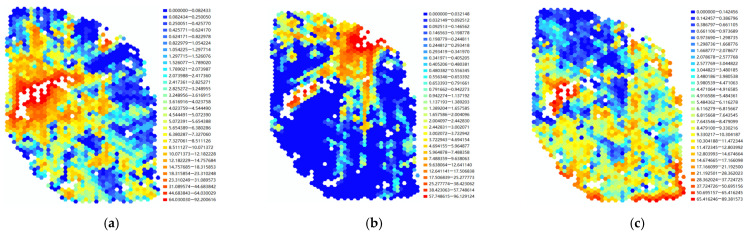
Correlated variables with the landscape visual value of the global calculation: (**a**) high obstruction area for landscape A; (**b**) high obstruction area for landscape B; (**c**) high obstruction area for landscape C; (**d**) spatial explanation of the variables, including overall visibility, density, Pro_Len, 90De_Angle, Dis_Pt, overall blocking of view, Geo_Len, and height.

**Figure 6 ijerph-20-01788-f006:**
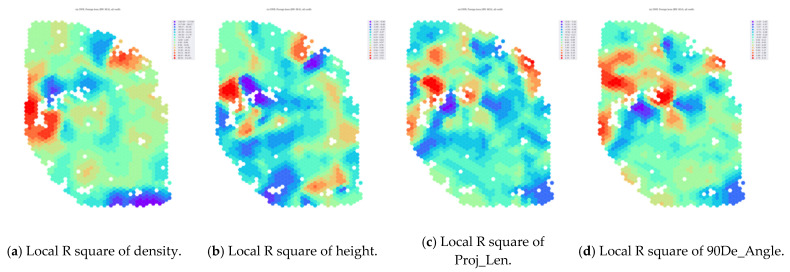
Influential areas and relative intervention methods: (**a**) density method; (**b**) height method; (**c**) projection length method; (**d**) orientation of the viewpoints direction method.

**Figure 7 ijerph-20-01788-f007:**
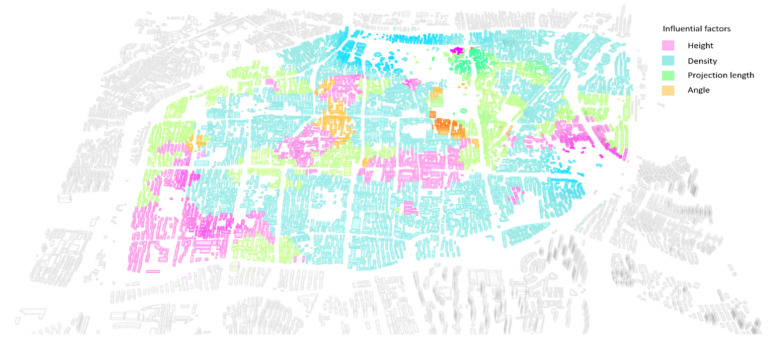
Influential areas and relative intervention methods.

**Table 1 ijerph-20-01788-t001:** Summarizing the main analytical methods that have evolved [25] and comparing and discussing the value of their evaluation and impact on the urban model in the design process.

Visibility Method	Authors	2D or 3D	Measuring the Visible Building Surface	Measuring the Volume of Obstacles	Responsiveness of Analysis–Design Interaction	Superposition of Multiple Analysis Objects	Embodiment of Complex Urban Saturations in the Case Test
2D Visibility	Amidon and Elsner [18]; Travis et al. [26]	2D	No	No	No	No	No
2D Isovist	Tandy [17]; Benedikt [19]; Batty [20]; Turner et al. [21]	2D	No	No	No	No	No
2D VGA	O’Sullivan and Turner [5]; Turner et al. [21]	2D	No	No	No	Not Tested	No
2D Viewshed	Felleman [27]; Van den Berg, Van Lith, and Roos [28]	2D	No	No	No	No	No
Fuzzy and Cumulative Viewshed	Fisher [29,30]; Ogburn [31]	2D	No	No	No	Not Tested	No
Visualscape	Llobera [22]	2D	No	No	No	Not Tested	No
Viewsphere	Yang, Putra, and Li [24]	3D	No	No	No	Not Tested	No
Viewshed on Voxel model	Pyysalo, Oksanen, and Sarjakoski [32]	3D	No	No	No	No	No
LoSe and Voxelized Scene	Hagstrom and Messinger [33]	3D	Not Tested	No	No	Not Tested	No
3D Isovist	Morello and Ratti [34]	3D	Not Tested	No	No	Not Tested	Yes
3D Isovist—Advanced Spatial Openness Index	Fisher-Gewirtzman [35,36]	3D	Yes	No	Not Tested	Not Tested	No
3D Isovist (Voxel Method)	Chmielewski [37]	3D	Yes	Not Tested	Not Tested	Not Tested	No
3D Isovist—Optimized by Remote Sensing and LiDAR	Kim, Kim, and Kim [38]; Zhao et al. [39]	3D	Yes	No	Not Tested	Not Tested	No
3D Visibility (Ray Casting)	Koltsova, Tuncer, and Schmitt [40]	3D	Yes	No	Not Tested	Not Tested	No
3D Vector Visibility	Suleiman, Joliveau, and Favier [41]	3D	Yes	No	Not Tested	Not Tested	No
3D Viewshed (Vertex Shader)	Feng et al. [42]	3D	Yes	No	Not Tested	No	No
3D Visibility (Voxel Method)	Chmielewski and Tompalski [25]	3D	Yes	Not Tested	Not Tested	Not Tested	No
3D Visibility (with Visual Experience)	Chang and Park [43]	3D	Yes	No	Not Tested	Not Tested	Yes
3D VGA	Varoudis et al. [44]	3D	Yes	No	Not Tested	Not Tested	No
3D Space Syntax	Morais et al. [45]; Ascensão et al. [46]	3D	Not Tested	No	Not Tested	Not Tested	Yes

**Table 4 ijerph-20-01788-t004:** Typical research and case studies.

Typical Research	Tools/Software	Input Data Categories	Analysis Resolution Space	Study Object Types	Study Area	Basic Evaluation Rules	Validation
Morais et al. [45]	Depth Space 3D	Buildings, Landform	10 m	Measured all buildings and ground surfaces	Casa da Música, approximately 0.8 km^2^	Using the visualization to show the different kinds of space densities	Open CV to trace space dynamic
Ascensão et al. [46]	Depth Space 3D	Buildings, Landforms, Vegetation	Not mentioned	Measured all buildings, trees, and ground surfaces	Maia Park, approximately 0.2 km^2^	Percentage of the visible pixels of the total pixels; seasonal changes in vegetation on visibility	-
Morello and Ratti [34]	3D Isovist, Optimized in ArcGIS	Buildings	Not mentioned	Measured all building geometries	Milan Trade Fair, 1 km^2^	Using the line-of-sight method to evaluate Kevin Lynch’s 5 attributes through movement	-
Suleiman, Joliveau, and Favier [41]	ArcGIS and MATLAB	Buildings	Not mentioned	Measured the visibility of the building facade from a vantage point	500 m radius area	Compute more open space indices in a 3D environment, such as openness, enclosure, and scale	Compared with the images of the 3D environment
Rodriguez Cervilla et al. [52]	ArcGIS	Landform	10 m	Measured the 3D viewshed of terrain from a vantage point	The coastal area of the city of Malaga, 400 km^2^	to quantify the Visible volume for a large number of observers in a digital elevation model	-
Yin [63]	ArcGIS and ArcScene	Buildings	Not mentioned	Line-of-sight analysis for the building skyline	Downtown Buffalo, 1 m long street	Measure the street-level urban design qualities	-
Chmielewski and Tompalski [25]	Voxel-based method	Buildings, Trees, and Terrain	10 m	Measured the visibility with voxels box through a large number of observation points	8.8 ha area	Measure the volume of the voxel box to optimize the location of the OA board	-
Chmielewski [37]	Voxel-based 3D Isovist method and ArcGIS pro	Buildings, Trees, and Terrain	0.25 m	Measured the volume of the voxel bounding box (i.e., 3D VGA), the shading of trees was also taken into account	Lublin City (Poland), 1 km radius area	Measure the volume of the voxel bounding box to evaluate the landscape openness	-
Koltsova, Tuncer, and Schmitt [40]	Grasshopper, Ray casting method	Buildings, Terrain, Road network	Not mentioned	Measured the visibility of the building facade	Not mentioned, two street block	Combined visibility and accessibility to analyze the visual change through movement	-
Tara, Belesky, and Ninsalam [51]	ArcGIS, Grasshopper	Buildings	4–90 m	Line-of-sight analysis for the building skyline and measured the height and distance of the sight	Unlimited public locations in Melbourne	Measure the urban quality through the sky ratio and fractal dimension	-
Puspitasari and Kwon [62]	Grasshopper Ladybug	Buildings, Landform	5 m	Only studied tower blocks with a height higher than 100 m	1–4 km^2^ (1.7 km^2^ in Lujiazui District)	Percentage of the visible surface area	-

**Table 5 ijerph-20-01788-t005:** Measurement indices [43].

Indices	Mathematical Expressions	Diagram	Indices	Mathematical Expressions	Diagram
1 Density	The density of buildings within a 100 m radius of the selected building unit	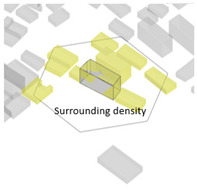	4 Angle deviation with 90 (90De_Angle)	90 degrees minus the angle between the long side of each building form’s envelope rectangle and the view direction	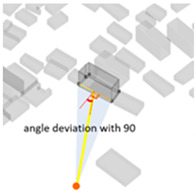
2 Height	The relative height of the building unit above ground level	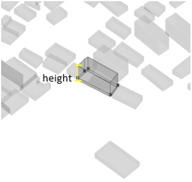	5 Geometries projection length (Proj_Len)	The actual projection length of the long side of the building’s envelope rectangle relative to the viewpoint	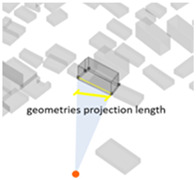
3 Geometries length (Geo_Len)	The perimeter of the base of each building unit	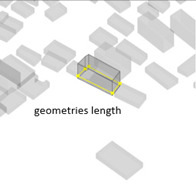	6 Distance from landscape focus point (Dis_Pt)	The linear distance of each building to the landscape focus point	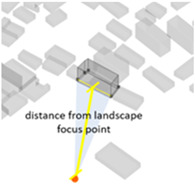
7 Visibility	The visible score on its surface, and the calculation was computed using Ladybug Grasshopper	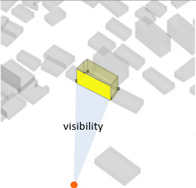	8 Visibility blocking	Detected the visibility interference of this building to the buildings behind its surrounding view direction [43]	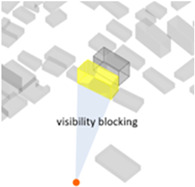

**Table 6 ijerph-20-01788-t006:** VIF test.

	Variable	VIF
0	density	8.026
1	Av_height	6.884
2	sum_height	8.098
3	sum_area	9.961
4	Av_ev_height	16.627
5	Av_pro_len	4.775
6	Av_90de_angle	7.04
7	All_dis_pt	4.071
8	All_Vis	6.856
9	All_block	3.032

**Table 7 ijerph-20-01788-t007:** OLS correlation of the selected variables.

Variable	coef	SE	t	*p*-Value	0.025	0.975
Density	−21.185	2.914	−7.269	0.000 ***	−26.902	−15.467
Height	0.338	0.055	6.185	0.000 ***	0.230	0.445
Proj_Len	0.737	0.173	4.259	0.000 ***	0.397	1.076
90De_Angle	0.283	0.100	2.842	0.000 ***	0.088	0.479

Notes: *** stand for significance level, *p* value < 0.01.

**Table 8 ijerph-20-01788-t008:** OLS and GWR correlation of the selected variables.

	OLS Method				GWR Method				
Variable	Est.	SE	t (Est/SE)	*p*-Value	Mean	STD	Min.	Median	Max.
Density	−20.602	2.635	7.819	0.000	0.419	30.793	−145.002	3.181	112.448
Height	0.263	0.049	5.360	0.000	0.280	0.448	−1.591	0.238	2.725
Proj_Len	0.736	0.150	4.915	0.000	0.499	1.273	−6.411	0.328	7.297
90De_Angle	0.310	0.088	3.521	0.000	0.166	0.741	−3.485	0.117	4.150

## Data Availability

The study did not report any publicly archived datasets.

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
