# Peer review of "A Survey of the Landscape Visibility Analysis Tools and Technical Improvements"

_ijerph, 2023, doi:10.3390/ijerph20031788_

Round 1

Reviewer 1 Report

This manuscript provided a comprehensive review of the development of landscape visibility analytical tools and their limitations. In this respect, the survey part of the manuscript is valuable for readers to acknowledge such tools. 

Unfortunately, I think the manuscript's primary shortcoming is the language. As for me, I found it difficult to understand the exact technical improvements that the authors have made. It seemed that the multiple (dynamical) viewpoints in the 3D dimensional space have already been (partially) implemented in previous tools/case studies. I would suggest the authors narrow down their exact improvements.

Besides, what are landscapes A, B, and C being discussed in Figure 5?

In summary, I think the manuscript's language and organization have much space for improvement before it can be considered for potential publication.  

Reviewer 2 Report

Dear authors ,

The article " A Survey of the Landscape Visibility Analysis Tools and the Technical Improvement " is a scientific work of high quality, understandable and clearly written. The subject of the article is in line with the theme of the journal.
It also appears to be well documented, with a detailed literature review and an analysis of the strengths and weaknesses of the most commonly used landscape visibility analysis tools
The methodology section is well described, with a detailed description of the conceptual framework for the spatial visibility analysis method proposed in this paper
The results of the tested method are explained and presented not only in terms of images, but also in terms of statistical correlation variables.
However, the conclusions section is much too long and contains information about the interpretation of the research results. Therefore, I recommend dividing this section into two parts: the discussion part, which contains the evaluation and analysis of the proposed method (section 5.1, 5.2, 5.3. partly 5.4.) and a section with the conclusions. The conclusion part of the article should be comprehensive and concise at the same time.
